# *Streptococcus thermophilus* Phages in Whey Derivatives: From Problem to Application in the Dairy Industry

**DOI:** 10.3390/v14040810

**Published:** 2022-04-14

**Authors:** Mariángeles Briggiler Marcó, Nicolás Machado, Andrea Quiberoni, Viviana Suárez

**Affiliations:** Instituto de Lactología Industrial (Universidad Nacional del Litoral, Consejo Nacional de Investigaciones Científicas y Técnicas), Facultad de Ingeniería Química, Santiago del Estero 2829, Santa Fe S3000AOM, Argentina; nicom514@hotmail.com (N.M.); aquibe@fiq.unl.edu.ar (A.Q.); vivisuar@fiq.unl.edu.ar (V.S.)

**Keywords:** dairy whey derivates, *Streptococcus thermophilus*, bacteriophage, phage characterization, thermal resistance

## Abstract

Fifteen samples of whey protein concentrate (WPC) were tested against 37 commercial *Streptococcus thermophilus* strains to detect infective bacteriophages. Seventy-three diverse phages were isolated from 12 samples, characterized by using DNA restriction patterns and host range analyses. Sixty-two of them were classified as *cos*, two as *pac*, and nine as 5093, according to PCR multiplex assays. Phage concentration was greater than 10^4^ PFU/g for 25.3% of isolated phages. Seven phages showed an unusual wide host range, being able to infect a high number of the tested strains. Regarding thermal resistance, *pac* phages were the most sensitive, followed by *cos* phages, those classified as 5093 being the most resistant. Treatments at 85 °C for 5 min in TMG buffer were necessary to completely inactivate all phages. Results demonstrated that the use, without control, of these whey derivatives as additives in dairy fermentations could be a threat because of the potential phage infection of starter strains. In this sense, these phages constitute a pool of new isolates used to improve the phage resistance of starter cultures applied today in the fermentative industry.

## 1. Introduction

The re-use of cheese whey became a common practice in recent years because of the development of diverse membrane technologies, which has allowed to recover and to give added value to this by-product of the dairy industry. In particular, the addition of whey proteins in the manufacture of new dairy products is aiming to increase the process yield, improving also the nutritional level and/or the texture of the product [1,2]. In this way, the whey protein concentrate (WPC), whey protein isolate (WPI) or whey protein hydrolysate (WPH), microparticulated whey protein, whey permeate (WP), whey cream, whey powder, lactose powder, are some products that can be obtained from cheese whey [1]. 

To obtain these powder derivatives, cheese whey is subjected to diverse consecutive steps such as thermal treatment, membrane filtration, and spray drying [1,3,4]. It is well known that the lactic acid bacteria (LAB) phages are able to survive pasteurization and even stronger thermal treatments applied to raw milk during its processing [3,5]. Phages surviving these treatments may propagate during fermentations, reaching high concentrations in the final product or by-products. In this sense, titers up to 10^9^ PFU/mL in cheese whey have been detected [6,7]. On the other hand, the membrane technology applied depends on the derivative type to be obtained. In the case of WPC (main derivative used as an additive in fermentative dairy processes), the methodology is mainly the ultrafiltration (UF) and the pore size of the membrane allows to retain the phages potentially present besides the proteins; thus, phage concentration could increase (up to 10 times) in the final product [3,8]. In this sense, phages infecting diverse species of lactic acid bacteria (LAB), have been isolated from whey powders, in concentrations that can reach 10^7^ PFU/g [4].

In Argentina, the WPCs that can potentially contain phage particles are added to fermentative processes (cheese or fermented milks) without any control. Consequently, the obtained products may not have the expected characteristics and, even worse, may not be microbiologically safe [3]. In this way, an ingredient aiming to improve the characteristics of a certain fermented dairy product can become a dangerous source of phages, leading to the failure of the entire production, with the consequent negative economic impact. Thus, by-products such as the WPC may constitute a new and important source of phages to fermentation lines [9]. Considering the growing use of these whey derivates in fermentative processes, a systematic monitoring (and isolation) of phages present in these products would be essential in order to characterize them.

In Argentina, *Streptococcus thermophilus* is, undoubtedly, the most widely used lactic acid bacteria to create fermented products in the dairy industry. It is part of the starter cultures for making yogurt, fermented milks, and diverse types of cheeses [10]. Particularly, the presence of specific phages of LAB in whey derivates have not yet been studied and evidenced in Argentina. As previously described, the main aim of this study was to detect, isolate, and characterize *St. thermophilus* phages from WPC samples, of diverse origins, used routinely in local dairies. Isolated phages constitute a pool of new candidates, currently under study, to be used as tools to improve the phage resistance of dairy starter cultures.

## 2. Materials and Methods

### 2.1. Bacterial Strains and Growth Conditions

In this work, 37 *Streptococcus thermophilus* strains, isolated from commercial starters used in Argentinian dairy plants and available at INLAIN Collection, were selected to test phage presence in WPC samples. These strains were routinely grown (37 °C) in M17 broth and in reconstituted dry skim milk (RSM; 10%, *w*/*v*), and stored as frozen stocks (−80 °C) in M17 broth (Biokar, Beauvais, France) in the presence of 15% (*v*/*v*) of glycerol.

### 2.2. Phage Isolation

A total of 15 WPC samples from one whey-processing plant (plant I) and two Argentinian dairies (dairy II and dairy III, Table 1) were assayed against the 37 strains of *St. thermophilus* previously selected. In particular, samples provided by plant I were obtained from a mixture of cheese whey of dairies from diverse Argentinian regions. Samples were reconstituted (10% *w*/*v*) in distilled water and centrifuged at 13,000× *g* for 15 min at 5 °C. Afterwards, supernatants were filtered through membranes of 0.45 µm pore size (GVS Filtration Inc., Findlay, OH, USA). Filtrates were conserved at 4 °C and assayed for phage detection using the turbidity test [11].

Overnight cultures of the strains were inoculated (2%, *v*/*v*) in tubes containing M17-Ca broth (M17 broth supplemented with CaCl_2_ in a final concentration of 10 mM) and 500 µL of each filtrate was added. The controls were tubes without addition of the filtrates. Tubes were incubated at 37 °C until adequate growth of the strains in the control tubes was observed. Three consecutive subcultures (2%, *v*/*v*) in M17-Ca broth in order to increase the concentration of the phages present in the samples were completed. Tubes exhibiting complete lysis were subjected to a double-layer plaque titration method to check the presence of lysis plaques [11]. Phage purification was carried out by taking an isolated lysis plaque and spreading the phage on the sensitive strain. Afterwards, purified phages were quantified (double-layer plaque titration method) and propagated using a multiplicity of infection (m.o.i. = number of phage particles/number of bacterial cells) of 0.01 in M17-Ca broth in order to obtain suspensions with high titers (10^8^–10^9^ PFU/mL). Propagated phages were filtrated by 0.45 µm membranes and kept as frozen (−80 °C) stocks in M17 broth supplemented with glycerol (15%, *v*/*v*) [12]. On the other hand, phage enumeration (plaque-forming units per gram, PFU/g) in the WPC samples was determined by the double-layer plaque titration method.

### 2.3. Phage Characterization

#### 2.3.1. Host Range

This assay was performed using the spot test [11]. Each purified phage (83) was tested against the 37 *St. thermophilus* strains included in this study. To carry out the assay, overnight cultures of the bacterial strains were mixed with M17 soft-agar (0.6%, *w*/*v*) and plated as a thin layer on plates containing M17-Ca agar (1.2%, *w*/*v*). Then, 30 µL aliquots of decimal dilutions of each phage suspension in high titer (10^8^–10^9^ PFU/mL) were spotted onto the plates. After incubation at 37 °C for 16–18 h, the presence or absence of lysis zones was observed [11].

#### 2.3.2. DNA Extraction and Restriction Enzyme Analysis

Phage DNA was extracted with phenol-chloroform and precipitated with isopropyl alcohol from 1 mL of phage suspensions in high concentrations (>10^8^ PFU/mL), according to Pujato et al. [13]. After washing with absolute ethanol, DNA pellets were resuspended in double-distilled and nuclease-free water (Gibco™, Invitrogen, Grand Island, NY, USA). Phage DNA was digested with EcoRV and EcoRI endonucleases according to the manufacturer’s recommendations (Genbiotech, Antibes, France). DNA fragments were resolved by agarose gel electrophoresis (0.8%, *w*/*v*) [14] and captured using the Kodak Electrophoresis Documentation and Analysis System 290 (EDAS 290, Celbio, Milan, Italy). Restriction patterns were analyzed with software package BioNumerics™ (version 6.0; Applied Maths BVBA, Sint-Martens-Latem, Belgium), using the unweighted pair group method with arithmetic mean (UPGMA) for phage grouping and Jaccard coefficient for profile discrimination [15].

#### 2.3.3. Determination of Genetic Group

A multiplex PCR reaction was performed to identify the genetic group to which the isolated phages belong. The last group (P738) recently reported [16] was not included in this determination. The oligonucleotides used and the expected product size for each genetic group are shown in Table 2. PCR conditions were: one cycle at 95 °C for 2 min followed by 30 cycles (95 °C for 15 s, 55 °C for 30 s, and 72 °C for 1 min) and a final cycle at 72 °C for 10 min [7]. The amplicons were resolved by agarose gel electrophoresis (1%, *w*/*v*) and photographed as was previously described.

#### 2.3.4. Thermal Resistance

A total of 17 phages in relation to their thermal resistance were studied. Their genetic group, infective capacity, and persistence in the samples were the selection criteria for these phages. The selected phages are: phages *pac* (M1-A2, M1-H2), 5093 (M14-G1b, M1-G1a, M1-G1b, M13-G1a, M13-G1b, M14-B2, M15-B2B), and *cos* with persistence in diverse samples (M9-A10, M9-A2) or with high virulence (M3-A8b, M7-D1c, M13-D1c, M13-A8b, M13-A11, M15-A6). Two 5093 phages with plaque formation difficulty under the titration condition were not included in this experiment. Assays were performed in glass tubes (inner diameter of 0.94 ± 0.02 cm) containing 1 mL of the phage suspension (concentration between 10^6^ and 10^8^ PFU/mL) in Tris Magnesium Gelatin buffer (TMG; 10 mM Tris-HCl, 10 mM MgSO_4_, and 0.1% *w*/*v* gelatin, pH 7.4). Experiments were conducted at 75, 80, and 85 °C for 5 min, in triplicate independent assays. Titers of the infective phages before and after each thermal treatment through double-layer plaque titration method were determined and results were expressed as reduction in the phage infectivity (log orders). Data were processed applying one-way ANOVA (Tukey´s test pos hoc) statistical treatment for each thermal treatment and genetic group. Additionally, a descriptive analysis using box-and-whisker was completed.

## 3. Results

### 3.1. Phage Isolation

In the present work, a total of 87 positive detections were observed (turbidity test). Specifically, 12 out of the 15 assayed WPC samples (80% of the total) were positive regarding the presence of phages. On the other hand, no cell lysis for samples M4, M11, and M12 on the 37 assayed *Streptococcus thermophilus* strains was observed. Between 1 and 3 lysed strains (phage detection) were obtained for 4 samples (33.3% of the positive samples) and between 4 and 8 lysed strains for another 5 of them (41.7%). In particular, for samples M13, M14, and M15, it was possible to detect infection on a high number of strains, that is 18 (48.6% of the 37 total strains), 15 (40.5%), and 17 (45.9%) strains, respectively (Table 1).

Regarding phage sensitivity of the tested *St. thermophilus* strains, 20 of them (54%) were sensitive to, at least, 1 sample whereas 17 strains (46%) were not lysed with any of the tested samples. Among those 20 strains, 9 strains (24.3%) lysed against 1-3 WPC samples, while another 9 strains showed to be sensitive against 4-7 samples. In particular, 2 strains (5.4%) showed high sensitivity to samples (cell lysis against 8–11 samples) (Figure 1). These strains, A4a (sensitive to 11 samples) and A8b (sensitive to 10 samples), belong to the same supplier (A) but dissimilar starter (starters 4 and 8, respectively). In some cases, strains belonging to the same starter (multiple strain starters, such as A4a and A4b; A8a and A8b) evidenced a dissimilar behavior. In fact, strains A4a and A8b showed high sensitivity to the samples since they were lysed by 11 and 10 WPC samples, respectively. On the contrary, no cell lysis for strains A4b and A8a against the 15 samples was detected.

### 3.2. Phage Concentration

Phage concentration in the WPC samples ranged between <1.0 × 10^2^ and 3.3 × 10^5^ PFU/g (Table 1). Specifically, 7 samples evidenced phage concentrations higher than 10^4^ PFU/g, a level considered of high risk to fermentative processes [3]. These samples, which were M1, M6, M8, M9, M13, M14, and M15, represent 58.3% of the positive samples (7 out of 12). For the remaining samples, concentration of phages ranged from <1.0 × 10^2^ to 5.0 × 10^3^ PFU/g (Table 1). On the other hand, 21.7% of the 87 positive detections showed concentrations <1.0 × 10^2^ PFU/g whereas 25.3% of them exhibited concentrations higher than 1.0 × 10^4^ PFU/g. Most detections (53.0%) showed concentrations varying between 1.0 × 10^2^ and 1.0 × 10^4^ PFU/g (data not shown).

From 87 detections, a total of 83 phages were purified and subjected to their characterization (host range and enzyme restriction patterns). The remaining 4 detections could not be propagated.

### 3.3. Phage Characterization

#### 3.3.1. Host Range

This assay was performed by means of the spot test. Figure 2 shows the percentual distribution of the phages in relation to the number of strains infected by each of them. Most phages (42.7%) could infect between 5 and 8 strains and 37.8% of them between 1 and 4 strains, including the control strain (strain with which phage was isolated) in both cases. Only 4 phages (4.9%) were not able to infect additional strains (only the control one). An elevated number of phages (11% of the total) infected a high number of strains (9–17). Lastly, a group of 7 phages (8.5%) demonstrated an unusual high infectivity since more than 18 commercial strains of *St. thermophilus* were lysed (and up to 30 strains) (Figure 2). These phages were isolated from samples M1 (2 phages), M3 (1 phage), M13 (3 phages), and M15 (1 phage).

On the other hand, the 37 strains were sensitive to, at least, 2 phages (up to 34 phages), considering the phage isolated when they were used as a control strain. Figure 3 shows the number of phages able to infect each of the tested strains. In particular, 12 strains (A2, A4a, A8b, A9, A10, B2, C1, D1c, F1a, G1a, G1b, and H2) showed a very high sensitivity (sensitive against 20–34 phages). Furthermore, 24.3% of the strains were sensitive against 1–6 phages, and 43.2% against 7–19 phages. In some cases, the strains belonging to the same starter showed different sensitivity against phages (A4a and A4b; A5a, A5b and A5c; A8a and A8b; A13a and A13b; B1a and B1b; D1a, D1b and D1c; F1a and F1b). In fact, strains A4a, A5a, A5b, A8b, A13b, B1a, D1c, and F1a showed high sensitivity, exhibiting lysis with 22, 7, 9, 31, 17, 15, 29, and 28 phages, respectively. On the contrary, the remaining strains of each starter evidenced lower sensitivity, being lysed by 8 (A4b), 3 (A5c), 10 (A8a), 8 (A13a), 3 (B1b), 4 (D1a), 2 (D1b), and 7 (F1b) phages, respectively. As can be seen in Figure 3, both strains isolated from G1 starter culture showed high sensitivity against the isolated phages.

#### 3.3.2. Restriction Enzyme Analysis

Restriction enzyme patterns were obtained for the 83 phages from which DNA could be extracted. Profiles obtained with EcoRV showed high diversity among all phages studied, isolated from diverse samples and origins (Figure 4). Among them, 23 phages (27%) showed a similarity percentage higher than 90% (taken as limit of discrimination). Therefore, these phages were also subjected to EcoRI restriction in order to improve their discrimination (data not shown).

Notoriously, high phage diversity was found among the phages isolated from samples M13, M14, and M15, for which the number of isolates was remarkably elevated. In fact, all phages isolated from samples M13 and M14 were different, whereas only two pairs of phages isolated from sample M15 were considered similar, taking as the limit a similarity percentage of 90 % (data not shown). Moreover, phages isolated from sample M14 were all different to those isolated from sample M15 (same origin for both samples) showing high phage diversity even between samples from the same origin (Figure 5).

Finally, taking into account the restriction enzyme patterns and the host range analysis, 73 out of the total of 83 phages studied were classified as different (87.9%) and subjected to analysis for their genetic group determination.

#### 3.3.3. Determination of Genetic Group

Most of the 73 phages were classified as *cos* (62 phages out of 73, 84.9% of the total), whereas 2 of them (M1-A2 and M1-H2) belong to the *pac* group (2.7%). On the other hand, 9 phages (M1-G1a, M1-G1b, M7-G1a, M13-G1a, M13-G1b, M14-B2, M14-G1b, M15-B2A, and M15-B2B) were classified as 5093 (12.4%). In particular, phages M7-G1a and M15-B2A evidenced also a weak band corresponding to the *cos* group. Figure 6 shows, as an example, the amplicons obtained for phages belonging to the three groups found.

#### 3.3.4. Thermal Resistance

Phages classified as *pac* (M1-H2 and M1-A2) were completely inactivated after 5 min at 75 °C, with count reductions of, at least, 5.8 log orders. Instead, phages belonging to groups *cos* and 5093 were slightly influenced by this treatment. Phage reductions were lower than 1 log order for *cos* phages, with values ranging between 0.40 ± 0.28 (phage M13-D1c) and 0.84 ± 0.03 (phage M13-A8b), except for phages M9-A10 (1.38 ± 0.46) and M15-A6 (1.47 ± 0.02). Regarding phages classified as 5093, decreases in phage counts were even smaller than for *cos* phages and they dropped between 0.07 ± 0.04 (M1-G1a) and 0.26 ± 0.31 (M13-G1a) log orders, except for phages M14-G1b (0.80 ± 0.11) and M13-G1b (1.29 ± 0.08) (Figure 7A and Table 3). Statistical analysis with one-way ANOVA (α = 0.05, Tukey’s test pos hoc) demonstrated a greater variability in the response to the treatment for *cos* phages (3 subgroups) in comparison to the 5093 ones (2 subgroups) (Table 3).

Similarly, for a thermal treatment at 80 °C for 5 min, phages belonging to the *cos* group showed higher thermal sensitivity than 5093 phages. Results showed that 3 *cos* phages (M9-A10, M9-A2, and M15-A6) were completely inactivated (Figure 7B) whereas, for the remaining *cos* phages, phage reductions ranging between 2.21 ± 0.35 (M13-A11) and 4.19 ± 0.24 (M3-A8b) log orders were obtained. On the other hand, none of the 7 phages of group 5093 lost their infectivity completely with this treatment. Phages belonging to this group diminished their counts between 0.28 ± 0.07 (M1-G1a) and 1.38 ± 0.51 (M14-G1b) log orders, with a higher phage reduction for M13-G1b phage (3.56 ± 0.06 log orders) (Figure 7B and Table 3). Higher variability in the response to the treatment was again observed for *cos* phages (4 subgroups) in comparison to the 5093 ones (2 subgroups).

Finally, none of the phages were able to resist 85 °C for 5 min; that is, no phage particles (<10 PFU/mL) were detected after this treatment.

In addition, the box-and-whisker plot method was used to analyze and compare the global dispersion of the thermal resistance values obtained for the phages belonging to each genetic group (*cos* and 5093), when analyzing both thermal treatments. The treatment at 75 °C for 5 min showed lower value dispersion for the *cos* phages in the second and third quartiles (50 % of phages) than for 5093 phages. However, a wide dispersion of the values for the rest of the *cos* phages (the other 50%) was observed (Figure 8A), even showing extreme ones in two cases. As observed, overall, the 5093 phages were more resistant than the *cos* group ones, as the median value was lower for that group.

On the contrary, for the treatment at 80 °C for 5 min, the *cos* phages showed greater response variability than the 5093 phages, with a normal distribution of the values (without extreme points) and, consequently, with a median value close to the mean value (Figure 8B). On the other hand, the response of 5093 phages was similar to that found for the previous treatment (75 °C for 5 min), with a median value close to the lowest end of the box and significantly lower than for *cos* phages.

Among thermal treatments, it clearly seems that the strongest temperature treatment caused a better discrimination in relation to the diversity of response, since the values are located in a greater range of reduction for both groups, when comparing to those obtained at 75 °C.

## 4. Discussion

*St. thermophilus* is a LAB species widely used around the world in fermented milk and cheese development. The indiscriminate use of WPC as an additive in the fermentative dairy industry can lead to severe consequences if active phages against the starter strains are present. The infective action of these phages could delay or stop the activity of these strains, resulting in significant economic losses.

The screening performed in this study showed that 80% of the assayed WPC samples (12 out of 15) contained infective phages when they were tested against 37 commercial strains of *St. thermophilus*. Wagner et al. [4] studied the presence of phages in 5 samples of whey formulations (WF) and 13 whey powder (WP), against strains of diverse LAB species used in cheese manufacture (*Lactococcus lactis*, 8 strains; *St. thermophilus*, 5 strains; *Leuconostoc pseudomesenteroides*, 4 strains; and *Leuconostoc mesenteroides*, 2 strains). Infective phages were detected in all WF and WP samples assayed. In particular, the authors isolated *St. thermophilus* phages from 1 out of the 5 WF samples (20%) and from 9 out of the 13 WP samples (69.2%), i.e., 55.5% of total samples.

It is important to remark that, in our work, for 3 of the tested samples (20% of the total), between 15 and 18 detections were observed, indicating that their use as additives in subsequent fermentative processes could be extremely risky. Specifically, the sample M13 (WPC 80), provided by a cheese whey-processing plant, was the most infective sample. These results are consistent with those expected, since M13 is a whey mixture of diverse origins and more concentrated in proteins (80%); thus, a greater phage detection would be anticipated. The other 2 samples, M14 and M15 (both WPC 35) that came from the same dairy plant, are used in their own production and, possibly, are commercialized to other smaller dairies. This crossing, without any type of control, undoubtedly exacerbates the problem of possible infections due to higher diversification of the phage population.

The viral load found in the positive samples was, at least, 10^3^ PFU/g (depending on the isolated phage), with the exception of M10 whose maximum concentration was 5 × 10^2^ PFU/g. Particularly, 21 out of the 83 isolates (25.3%) showed titers higher than 10^4^ PFU/g. This value is a limit that marks a high risk for fermentation [19,20]. Consistently, Wagner et al. [4] reported phage levels between 10 and 10^5^ PFU/g in the studied WF and WP samples. These phage concentrations in powder whey derivatives are related to the high titers in which these populations can be present in the cheese whey [7,21,22], which have been found in levels between 10^2^ and 10^9^ PFU/mL. Additionally, the presence of phages in powder samples would be related to the intrinsic resistance of the virions to stress factors involved during the preparation process of whey derivatives (temperature and drying).

The phage diversity demonstrated in this study was extraordinarily wide, being of approx. 90% (73 out of 83 total phages), even amongst isolates from the same sample. These results agree with those found by McDonnell et al. [18], which reported the characterization of 40 *St. thermophilus* phages, isolated from whey samples from diverse geographical locations (Europe, USA, and Asia) and collected in different time periods. Although these authors do not directly show the results of all the phages isolated in their study, they affirm that the diversity found was very high. Similar results were reported by Lavelle et al. [7], who studied the diversity of 100 phages isolated from samples from different countries in Europe, Africa, and America. They remark that the phage diversity found could be a consequence of the genetic and phenotypic diversity of starters provided by diverse suppliers. In fact, phages belonging to the 4 known groups until that moment (*cos*, *pac*, 5093, and 987) were isolated [7]. It is noteworthy that the high diversity reported in these studies could be attributed to the fact that the samples came from different countries; while, in our work, the samples came, at most, from different regions of the country and, in some cases, even from the same dairy plant.

Regarding the genetic group classification, most of the phages isolated in this work belonged to the *cos* group (62 phages), followed by the 5093 group (9) and only 2 phages were classified as *pac*; whereas it was not possible to find phages belonging to the 987 group. The predominance of *cos* phages over the other groups is widely reported [4,7,17,18,23]. On the other hand, the 5093 [24] and 987 [25] groups were recently identified and, therefore, the awaited isolation frequency would be lower. However, in our study, a higher number of phages corresponding to the 5093 group than those to the *pac* one, were isolated. On the other hand, amongst the 9 phages of the 5093 group, some of them also exhibited the amplicon characteristic of *cos* phages. This fact leads us to think that these phages could constitute a new group, a hypothesis that must be verified by phage sequencing, comparing the degree of homology with the referential exponents of the other groups.

The host profile is a fundamental assay in the characterization of phages, because it provides direct information concerning the infection capacity of the virions. The results obtained in our work showed that most of the isolated phages (48) infected up to 5 strains of the 37 tested ones, while only 4 were capable of attacking only their respective control strain. Extraordinarily, 7 out of the total phages lysed a large number of strains (more than 18). All these results do not clearly agree with those reported by other authors so far. In effect, Binetti et al. [26] demonstrated that *St. thermophilus* phages isolated from yogurt and cheese production plants had a narrow host range. The authors tested 15 phages with 14 strains, obtaining only 4 phages that lysed a maximum of 4 strains (the same strains for the 4 phages). In the same sense, Mc Donnell et al. [18] determined the host range of 40 phages, isolated from cheese whey samples obtained from diverse dairy plants and different geographical points in a period between 2006 and 2012, against 91 strains of *St. thermophilus*. Most phages lysed only 1 additional strain to the host with which it had been isolated, and only a few of them were capable of infecting between 3 and 6 strains. On the other hand, Lavelle et al. [7] reported that, of 100 *St. thermophilus* phages assayed against 52 commercial strains, only 4 were capable of lysing more than 4 strains.

An interesting analysis to be carried out is the relationship between the genetic group and the infective capacity of the isolated phages. In our work, the two isolated *pac* phages (M1-A2 and M1-H2) belonged to the group of the 7 phages with the widest host profile. The relationship between genetic group and infective capacity was previously reported by Lavelle et al. [7]. As previously mentioned, the authors evaluated 100 phages, of which 79 were classified as *cos*, 9 as *pac*, 8 belonged to the 987 group, and 4 to the 5093 group. Out of 4 phages (out of the 100 studied phages) that were able to lyse more than 4 strains, 3 belonged to the *pac* group. These results indicate that 33% of the *pac* phages (out of 9 total) showed a broader range of host strains than the remaining groups, since only 1 *cos* phage (out of the 79 *cos* phages evaluated), could infect more than 4 strains. In our study, these would be very preliminary conclusions, since only 2 *pac* phages were isolated and studied, which is a very limited number. Additional studies relating host range with genetic group should be carried out. On the contrary, different results were reported in the work of Binetti et al. [26], in which 16 commercial strains of *St. thermophilus* were evaluated against 17 phages belonging to the 2 genetic groups, *cos* and *pac*, known up to that moment, of which 2 phages (P13.2 and Sfi11) were classified as *pac*. Specifically, phage Sfi11 was unable to lyse any commercial strain, while only 1 strain was infected in the case of the phage P13.2.

Regarding *cos* phages isolated in our work, most of them infected between 2 and 10 strains, including the control one. Within this genetic group, the phages M3-A8b, M7-D1c, M13-D1c, M13-A8b, M13-A11, and M15-A6 stood out, since 20, 14, 20, 18, 23, and 30 sensitive strains, respectively, were detected. As can be seen, 6 of the 62 (approx. 10%) *cos* phages showed an atypical high infectivity, not previously reported. As already mentioned, in the study carried out by Lavelle et al. [7], the investigated 79 *cos* phages showed more narrow host profiles.

The 5093 group was recently identified and, for this reason, significantly lower numbers of isolates have been reported. The 9 phages investigated in this work showed similar host profile (comparable to most *cos* phages), infecting between 2 and 6 strains. Lavelle et al. [7] reported that phages from this group (as well as from the 987 group) infected only one strain highlighting also that, at high phage titers, they were able to cross-infect the host strain of the 987 phages.

As known, diverse control strategies are used in dairy plants to minimize the negative effect of phage infections and, among them, thermal inactivation has been the most studied. Diverse studies have evaluated the thermal resistance of LAB phages, applying combinations of temperatures and times traditionally used for pasteurization (63 °C for 30 min or 72 °C for 15 s), treatments used in the manufacture of fermented milk (80 °C for 30 min or 95 °C for 10 min) and conditions recommended by the International Dairy Federation (FIL) (90 °C for 15 min), to ensure the complete elimination of phages [11]. In particular, for *St. thermophilus* phages, Binetti and Reinheimer [27] studied the thermal resistance (63, 72, and 90 °C for 45 min) of 5 phages in diverse media (TMG buffer, RSM, and enriched tryptic soy (ETS) broth), starting from a concentration of 10^6^ PFU/mL. None of the phages tested were completely inactivated at 63 °C, in any of the suspension media assayed. On the other hand, only 1 phage was totally eliminated after 30 min at 72 °C in TMG buffer. At 90 °C, none of the phages survived the 5 min exposure, in any of the suspension media. Furthermore, Capra et al. [28] reported the thermal resistance shown by 4 *St. thermophilus* phages, in RSM when they were exposed to 90 °C for 45 min and for an initial concentration of 10^7^ PFU/mL. The results showed that phage 031-D was the most sensitive, reaching complete inactivation after 10 min; whereas the other 3 phages needed 40 min to achieve the same result. On the other hand, Wagner et al. [29] studied the thermal resistance of 9 *St. thermophilus* phages, isolated from samples of whey derivatives, all belonging to the *cos* genetic group. This study was carried out starting from initial concentrations between 10^5^ and 10^7^ PFU/mL, in milk, in a temperature range between 50 and 85 °C. Results obtained were different from those reported by Capra et al. [28] since no phages survived 1 min at 85 °C. Thermal resistance evaluated in our work was performed at 75, 80, and 85 °C during 5 min in TMG buffer. The results were related with the genetic group, an assessment not previously reported and which is very interesting from the point of view of explaining the differences in the isolation frequency of phage groups. The 2 *pac* phages were completely inactivated with a treatment at 75 °C for 5 min, whereas the *cos* and 5093 phages were more resistant, requiring temperatures of 85 °C for the same exposure time for complete inactivation. In turn, the phages belonging to the group 5093 showed greater thermal resistance than those of the *cos* group, since the mean values of the decrease in counts, before and after the treatments, were lower for that group. Indeed, 3 *cos* phages were completely inactivated after exposition at 80 °C for 5 min, and the remaining 5 decreased their concentration by a higher level than phages of the 5093 group. These results could explain the high number of isolated 5093 phages and, although the *cos* phages constitute even the majority group, the 5093 group, more recently identified, may become the most isolated group in the future due to natural selection by temperature. However, the conclusions reached in relation to the behavior of the 5093 phages against different thermal treatments should be confirmed, since the studies were carried out on a limited number of phages compared to the *cos* group, whose phages were selected from a much larger number of isolates. In any case, this relationship, although preliminary, contributes strongly to the characterization of the *St. thermophilus* phages. On the other hand, it has been observed that the suspension medium could exert a protective effect of the phage particles towards high temperatures [5]. Apparently, it would be the protein-rich matrices that would provide greater resistance to heat. This effect was carefully studied by Atamer et al. [30], for 2 *L. lactis* phages (P008 and P680), showing that media containing lactic proteins (whey or casein) increased heat resistance, whereas fat had no influence. On the contrary, resistance of the phage particles during the thermal treatment decreased in the presence of salts or carbohydrates. In view of the protective effect of whey proteins and considering these derivatives as new ingredients in the production of fermented dairy products, Geagea et al. [9] selected the highly heat-resistant *L. lactis* phage P1532 to study the influence of WPC and individual whey components as lactoglobulin, lactalbumin, and bovine serum albumin under diverse pH and heat treatment conditions. This work confirmed that the protective effect of WPC is not restricted to one type of protein but to the combination of the 3 proteins studied. It should be noted that our study was carried out in buffer, so it would be expected that the heat resistance shown by our phages would be increased when thermal treatments are conducted in the presence of a milk matrix, as is undertaken in the industry.

## 5. Conclusions

Results obtained in this work demonstrated that WPC samples used by local dairy industries constitute a source of *St. thermophilus* phages with high diversity. Therefore, it should be considered when WPC is used as an additive in the manufacture of dairy fermentative products in order to avoid problems associated with phage infections. Moreover, and as an extension of the present research, the pool of isolated phages is currently under deeper characterization, in order to be used as tools to improve the phage resistance of starter cultures (through CRISPR-Cas technology); thus contributing to diminishing the frequency of infections in dairies.

## Figures and Tables

**Figure 1 viruses-14-00810-f001:**
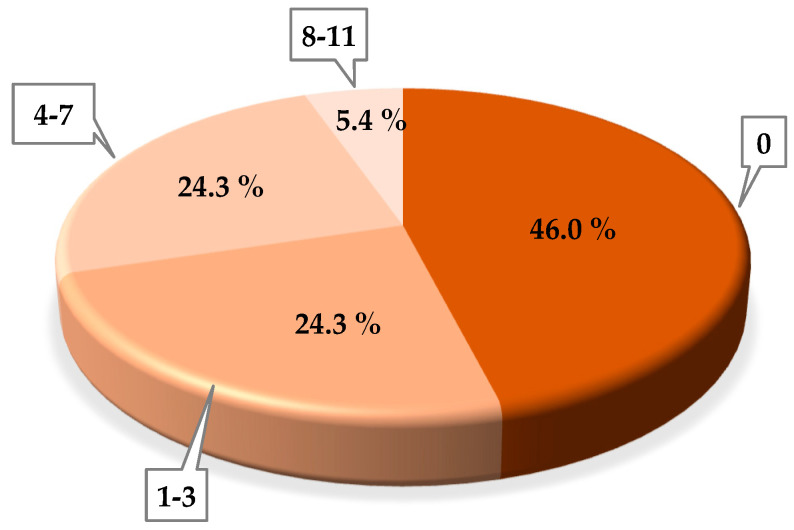
Percentual distribution of positive *St. thermophilus* strains according to their sensitivity against WPC samples (number of samples included in the bubble).

**Figure 2 viruses-14-00810-f002:**
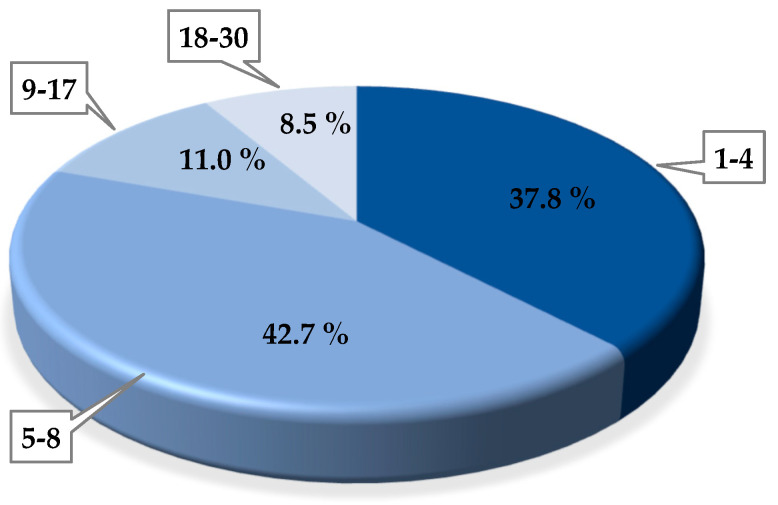
Percentual distribution of phages isolated from WPC samples considering the number of strains (in bubble) infected by each of them (infective capacity).

**Figure 3 viruses-14-00810-f003:**
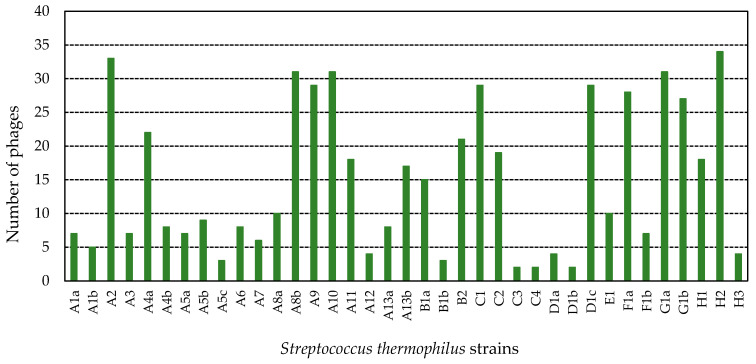
Sensitivity of 37 tested *St. thermophilus* strains against phages isolated from WPC samples, considering number of phages able to infect each of them. Uppercase letter indicates the supplier of the starter; number, the name of starter; and lowercase letter, the strain isolated from a starter.

**Figure 4 viruses-14-00810-f004:**
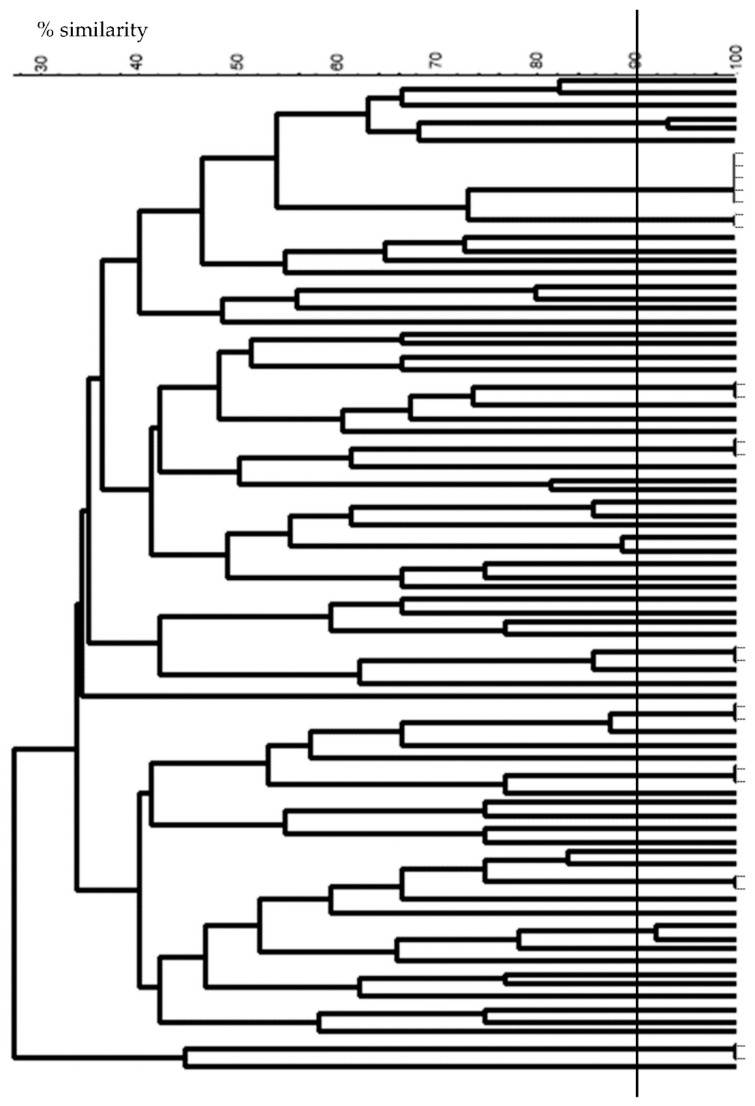
Dendrogram of restriction profiles obtained for 83 isolated phages from WPC samples, using EcoRV restriction enzyme (UPGMA and Jaccard factor).

**Figure 5 viruses-14-00810-f005:**
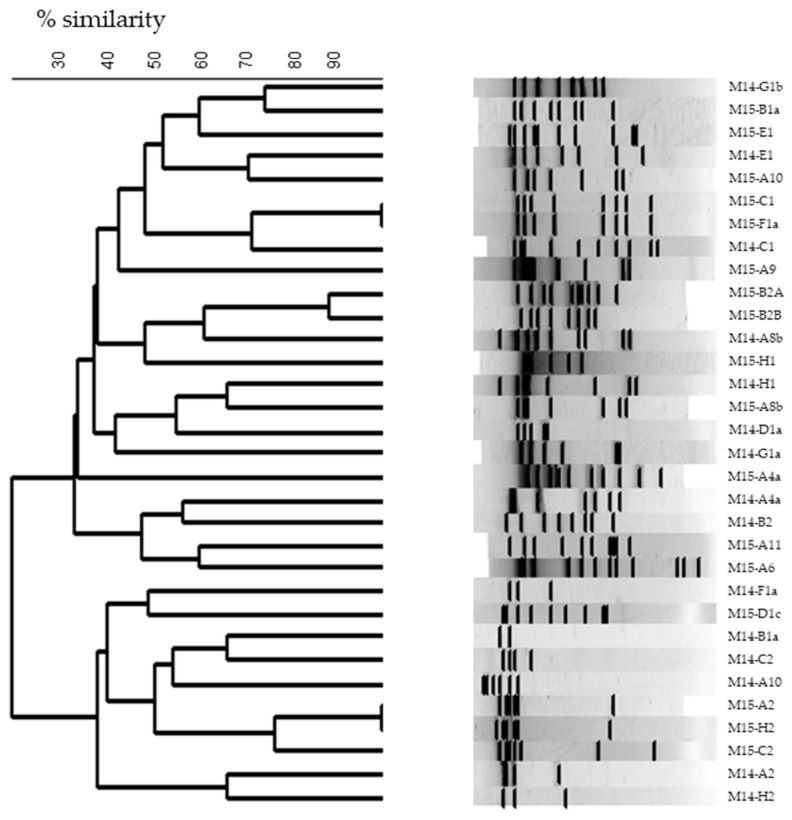
Dendrogram of restriction profiles obtained for isolated phages from M14 and M15 WPC samples, using EcoRV restriction enzyme (UPGMA and Jaccard factor).

**Figure 6 viruses-14-00810-f006:**
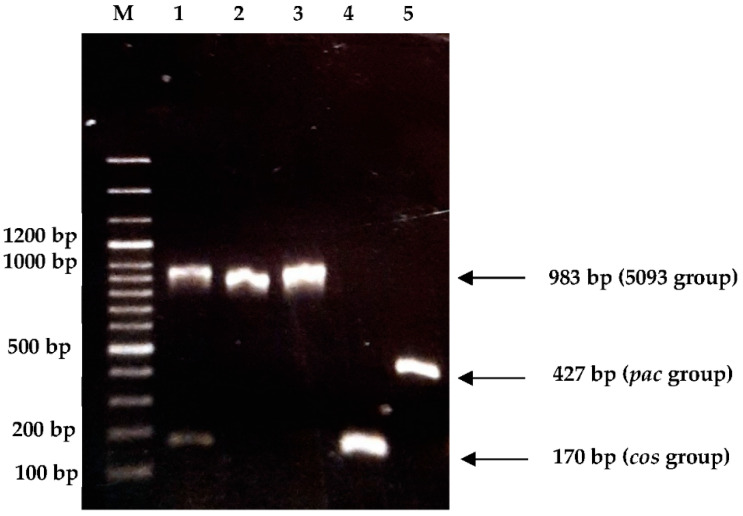
Agarose gel electrophoresis of amplicons obtained (multiplex typing) for different genetic groups of *St. thermophilus* phages. M: 100 bp Ladder plus, Dongsheng Biotech; 1, 2, 3, 4, 5: M15-B2 (A), M15-B2 (B), M14-G1b, M15-H2, and M1-A2 phages.

**Figure 7 viruses-14-00810-f007:**
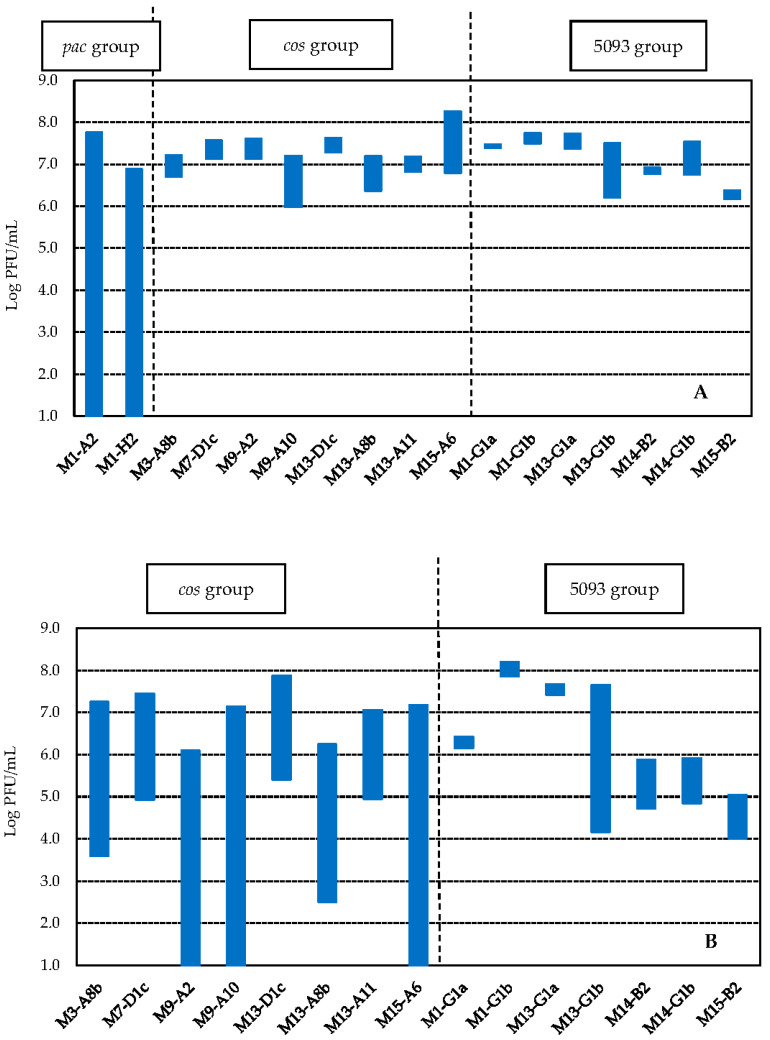
Inactivation of *St. thermophilus* phages, belonging to diverse genetic groups, after thermal treatments of 75 °C for 5 min (**A**) and 80 °C for 5 min (**B**). Bars indicate phage titers (log PFU/mL) before (**top**) and after (**bottom**) the thermal treatment applied.

**Figure 8 viruses-14-00810-f008:**
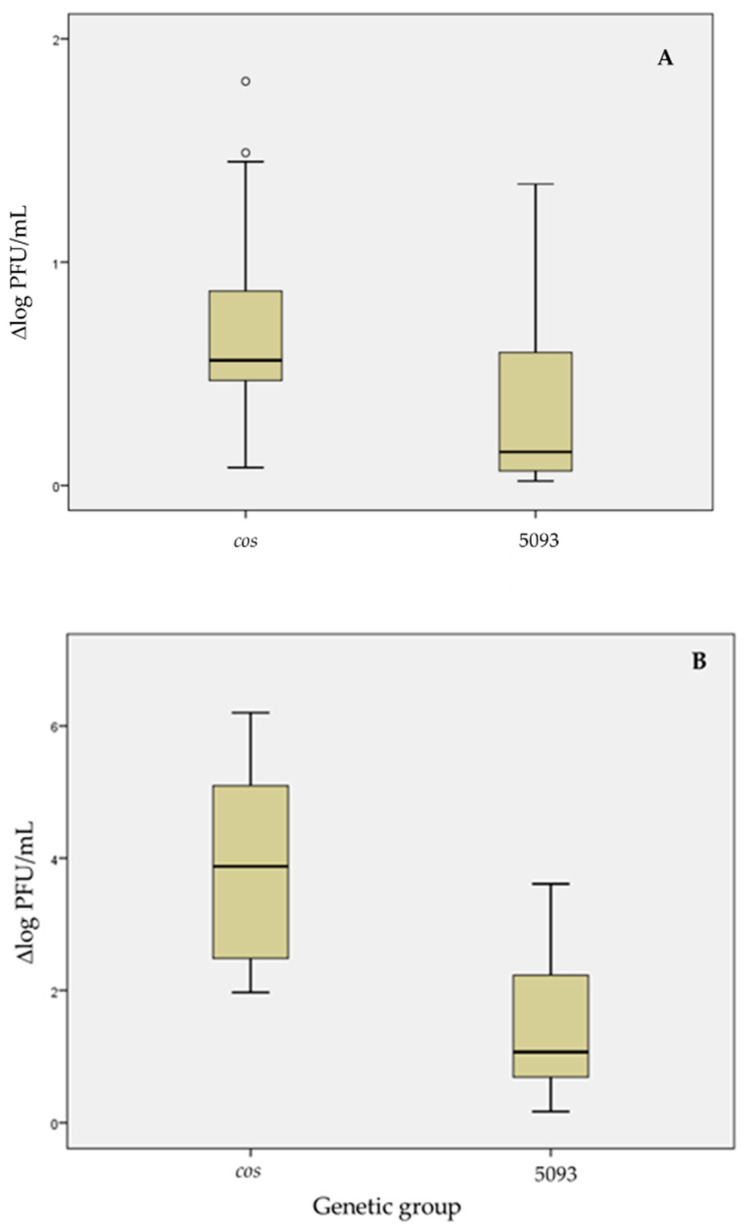
Box-and-whisker plots obtained for responses (Δlog PFU/mL, before and after thermal treatment) of diverse genetic group phages, at 75 °C for 5 min (**A**) and 80 °C for 5 min (**B**).

**Table 1 viruses-14-00810-t001:** Sample identification, number of lysed strains, and phage concentration in the positive samples.

Sample	Type ^d^	Lysed *Streptococcus thermophilus* Strains (Number)	Range of Phage Concentration (PFU/g)
M1^a^	WPC 35	5	<1.0 × 10^2^–2.2 × 10^4^
M2 ^a^	WPC 35	3	<1.0 × 10^2^–1.0 × 10^3^
M3 ^a^	WPC 35	4	<1.0 × 10^2^–4.3 × 10^3^
M4 ^a^	WPC 35	0	nd
M5 ^a^	WPC 35	1	5.0 × 10^3^
M6 ^a^	WPC 35	2	3.5 × 10^3^–2.6 × 10^4^
M7 ^a^	WPC 35	4	<1.0 × 10^2^–1.5 × 10^3^
M8 ^a^	WPC 35	8	<1.0 × 10^2^–1.6 × 10^5^
M9 ^a^	WPC 35	7	<1.0 × 10^2^–3.3 × 10^5^
M10 ^a^	WPC 35	3	1.0 × 10^2^–5.0 × 10^2^
M11 ^a^	WPC 35	0	nd
M12 ^b^	WPC 50	0	nd
M13 ^a^	WPC 80	18	<1.0 × 10^2^–1.1 × 10^5^
M14 ^c^	WPC 35	15	<1.0 × 10^2^–7.0 × 10^4^
M15 ^c^	WPC 35	17	<1.0 × 10^2^–1.9 × 10^4^

Samples provided by whey-processing plant I (^a^), dairy II (^b^), and dairy III (^c^). ^d^ Protein percentage in the WPC (whey protein concentrate) sample (35, 50, and 80 %). nd: not determined.

**Table 2 viruses-14-00810-t002:** Oligonucleotides used in multiplex PCR reaction and the expected product size for each phage genetic group.

Genetic Group	Primer Name	Sequence	Expected Size	Reference
*cos*	*cos* FOR	5′-ggttcacgtgtttatgaaaaatgg-3′	170 bp	[17]
	*cos* REV	5′-agcagaatcagcaagcaagctgtt-3′		
*pac*	*pac* FOR	5′-gaagctatgcgtatgcaagt-3′	427 bp	[17]
	*pac* REV	5′-ttagggataagagtcaagtg-3′		
5093	5093 FOR	5′-ctggctcttggtggtcttgc-3′	983 bp	[18]
	5093 REV	5′-gcggcaaccatcttagaccag-3′		
987	987 FOR	5′-ctaagcgtttgccactgtcag-3′	707 bp	[18]
	987 REV	5′-gctgccgttgtttgaaaac-3′		

**Table 3 viruses-14-00810-t003:** Resistance of selected *St. thermophilus* phages against diverse thermal treatments.

Genetic Group	Δlog PFU/mL * for Thermal Treatment ($\bar{x}$ ± δ):	
	75 °C for 5 min **	80 °C for 5 min **
** *cos* **		
M3-A8b	0.53 ± 0.04 ^a,b^	3.67 ± 0.09 ^b^
M7-D1c	0.44 ± 0.02 ^a^	2.52 ± 0.11 ^a^
M9-A2	0.41 ± 0.34 ^a^	5.10 ± 0.02 ^c^
M9-A10	1.38 ± 0.46 ^b,c^	5.22 ± 0.30 ^c^
M13-D1c	0.40 ± 0.28 ^a^	2.47 ± 0.06 ^a^
M13-A8b	0.84 ± 0.03 ^a,b,c^	4.19 ± 0.24 ^b^
M13-A11	0.47 ± 0.19 ^a^	2.21 ± 0.35 ^a^
M15-A6	1.47 ± 0.02 ^c^	6.18 ± 0.04 ^d^
**5093**		
M1-G1a	0.07 ± 0.04 ^a^	0.28 ± 0.07 ^a^
M1-G1b	0.24 ± 0.13 ^a^	0.54 ± 0.23 ^a^
M13-G1a	0.26 ± 0.31 ^a^	1.29 ± 1.22 ^a^
M13-G1b	1.29 ± 0.08 ^b^	3.56 ± 0.06 ^b^
M14-B2	0.09 ± 0.09 ^a^	0.86 ± 0.26 ^a^
M14-G1b	0.80 ± 0.11 ^b^	1.38 ± 0.51 ^a^
M15-B2(B)	0.14 ± 0.14 ^a^	1.12 ± 0.14 ^a^

* Difference in log PFU/mL before and after thermal treatment. ** Assays were performed in TMG buffer. Lowercase letters in the same genetic group and thermal treatment indicate significant difference between the mean values (one-way ANOVA, α = 0.05, Tukey *post hoc* test).

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
