# Peer review of "Streptococcus thermophilus Phages in Whey Derivatives: From Problem to Application in the Dairy Industry"

_viruses, 2022, doi:10.3390/v14040810_

Round 1

Reviewer 1 Report

The results of this work evidenced a high incidence of S. thermophilus phages in cheese whey derivatives, as a growing problem in the dairy industry, actually aggravated by the lack of adequate treatments and/or controls before the reuse of these whey batches in dairy production lines.

Although the subject has been previously addressed by other researchers, the present study highlights the surprisingly high occurrence of WPC samples with not only one but several, quite diverse infective phages. A high number of phages were isolated and correctly characterized to ensure that several phages from the same sample were different. Besides, a good analysis of number of infected samples, number and frequency of phages occurrence, number of strains infected by a different number of phages, etc., was conducted and discussed.

Some very interesting arose from the study, for example the high correlation between phage genetic group and thermal resistance, which has been correctly discussed. The isolation of new 5093 phages is also interesting.

The discussion section is organized and complete; however, it can be shortened, also reducing the use of series of names and numbers, which can be counterproductive in some way, making difficult for the reader to focus on the main global findings.

As stated in the conclusions, and considering the great pool of phage-strain interactions studied, it would be interesting to conduce a future characterization of the strains used this study, specially regarding phage resistant mechanisms.

English may be improved for clarity. Also, many grammar mistakes should be corrected.

Minor comments:

L47 and 63 - replace "our country" with "Argentina" and "in local dairies" respectively

L75: dairy II and dairy III (in concordance with Table 1)

L88: how do the authors know the moi? did they count phage particles in the lysis plaques? please explain

L79, L100 (and other places in the MS) - replace with "turbidity test", "spot test" (why uppercase?)

Table 1 - most ranges of titres begin with "<", why? were some phages detected but not quantified? and why not, if that was the case?

Section 2.3.3. - in my opinion, this information (at least primer name, sequence, expected size) could be better read in a table

L132 - please re write this sentence

Fig 7 - were thermal resistance tested in triplicate? in than case, error bars should be shown in the figure

Fig 8 - do not use decimals in x-axis labels (I would prefer using the phage group names directly in the axis, without double reference). Also, it would be better putting figs. 8A and 8B side by side, with the same y-scale, for best comparison between treatments

Author Response

Response to Reviewer 1

The results of this work evidenced a high incidence of S. thermophilus phages in cheese whey derivatives, as a growing problem in the dairy industry, actually aggravated by the lack of adequate treatments and/or controls before the reuse of these whey batches in dairy production lines.

Although the subject has been previously addressed by other researchers, the present study highlights the surprisingly high occurrence of WPC samples with not only one but several, quite diverse infective phages. A high number of phages were isolated and correctly characterized to ensure that several phages from the same sample were different. Besides, a good analysis of number of infected samples, number and frequency of phages occurrence, number of strains infected by a different number of phages, etc., was conducted and discussed.

Some very interesting arose from the study, for example the high correlation between phage genetic group and thermal resistance, which has been correctly discussed. The isolation of new 5093 phages is also interesting.

The discussion section is organized and complete; however, it can be shortened, also reducing the use of series of names and numbers, which can be counterproductive in some way, making difficult for the reader to focus on the main global findings.

As stated in the conclusions, and considering the great pool of phage-strain interactions studied, it would be interesting to conduce a future characterization of the strains used this study, specially regarding phage resistant mechanisms.

English may be improved for clarity. Also, many grammar mistakes should be corrected.

Minor comments:

L47 and 63: replace "our country" with "Argentina" and "in local dairies" respectively.

Changes were made in the manuscript following the reviewer’s suggestions.

L75: dairy II and dairy III (in concordance with Table 1).

We are in agreement with the reviewer’s comment. Changes were made in the manuscript.

L88: how do the authors know the moi? did they count phage particles in the lysis plaques? please explain.

Phage purification by taking an isolated lysis plaque was performed. Then, it was spread on the sensitive strain until complete cell lysis. Afterwards, phage enumeration by double-layer plaque titration method was determined. Thus, we were able to calculate the m.o.i. for the subsequent phage propagation.

It was clarified in the methodology (please, see lines 88-92).

L79, L100 (and other places in the MS) - replace with "turbidity test", "spot test" (why uppercase?)

We agree with the reviewer’s comment. Changes were done in the manuscript.

Table 1 - most ranges of titres begin with "<", why? were some phages detected but not quantified? and why not, if that was the case?

Titers < 1.0x102 PFU/g correspond to phages whose concentrations in the WPC samples were impossible to quantify because they were lower than the detection limit of the counting method (double-layer plaque method). In these cases, a titer (through double-layer plaque titration method) of a lysed tube from turbidity test was done and a lysis plaque was taken for phage isolation and purification.

Section 2.3.3. - in my opinion, this information (at least primer name, sequence, expected size) could be better read in a table

A table was included in the manuscript, as it was suggested by the reviewer.

L132 - please re write this sentence

Sentence was rewritten.

Fig 7 - were thermal resistance tested in triplicate? in than case, error bars should be shown in the figure.

Thermal resistance assays were performed in triplicate. Figures 7 with (and without) error bars were included in the manuscript. We think it would look better if figure 7 without error bars is included. Standard deviations of titer reductions caused by thermal treatment are included in Table 3. Maybe it is more interesting to show standard deviations of titer reductions than those for initial and final titers.

Fig 8 - do not use decimals in x-axis labels (I would prefer using the phage group names directly in the axis, without double reference). Also, it would be better putting figs. 8A and 8B side by side, with the same y-scale, for best comparison between treatments

The X-reference axis was changed as suggested. Regarding the scales of the Y axis, we prefer to keep the original ones, since the objective of these graphs is to compare behaviors between groups but not between treatments. We believe that, for this purpose, it is better to keep the original scales to analyze the dispersion in a better way.

Text about Fig. 8 in both results and discussion sections was modified (please, see lines 322-339; lines 491-496).

Reviewer 2 Report

Review of manuscript viruses-1652413

The work describes the identification and basic characterization of Streptococcus thermophilus phages detected in whey protein concentrates. In a simple experimental approach, the Authors show that WPC can be an important cause of Streptococcus thermophilus phage infections in the Argentinian dairy environments. In an elegant way it is argued that whey protein concentrates used as additive in dairy fermentation may be rich in phage content. The work presents data on the infective properties and thermal resistance of 73 newly isolated Strep. thermophilus phages. By such approach the Authors aim to advance the description of the properties of this group of phages.

However, the manuscript demands some modifications and language edition.

  1. The introduction is concise and gives a nice overview on the use of whey derivatives in dairy fermentations and the consequences of their uncontrolled application. A minor comment to this part of the manuscript is that one could think of elaborating in more detail on the applied membrane technology in respect to ‘derivative type to be obtained’. Also to my knowledge microfiltration is also used besides ultrafiltration.
  2. The materials and methods section lacks several information that would make the work plan easier to follow. The methodology is unclear at times as to how particular experiments were performed and on what type of samples (e.g., WPC or distinct phages) and by which method.
  3. The results section launches into phage isolation from WPC and goes on to characterize their host range – but this was done before restriction pattern analyses which confirm the distinctness of each phage isolate. Should not it be that the Authors would first confirm that they are dealing with distinct, purified phages before analyzing their host range?? The Authors chose to refer to phage and strain names directly in the text rather than in tables – this does not allow to quickly associate the data to a specific phage/strain and makes the reader back-tracked at times (e.g., l162-164).
  4. Discussion section relates the results of this study with literature data in respect to thermal resistance and host range and overall phage diversity among the isolates. However, some of the conclusions are not supported, while other aspects seem not to sufficiently discussed, e.g., it would add value to the publication to correlate the aspect of host range with genetic group, propose an explanation as to why no phages were detected in three WPCs all deriving from one source (dairy II), etc. It would be tempting also to try to correlate the protein % in WPC with the type of phages detected, also if possible it would be valuable to add an information on the genetic type of the detected phages in each WPC (i.e. in Table 1).
  5. The conclusions mention the long-term aim of this work, being application of these phages to increase phage resistance of starter cultures. This is also stated in the title (application). However, little is written to describe what approach is envisioned – although one can imagine how this can be done, it still seems that this issue is described very vaguely, especially for it to be mentioned in the title of this manuscript.

Overall, the importance of the presented data has not been communicated in a manner that allows the reader to identify it and properly interpret and should be revised. Despite these major concerns there is lots of good things included in the manuscript that advance the characterization of Str. thermophilus phages. It is clear that much effort was put into production of this data which will contribute to a better understanding of the biology of these phages and their persistence in the dairy environment.

Specific comments

L14-15: “All phages survived to pasteurization treatments commonly applied in dairy industry being necessary, at least, 85 °C for 5 min to inactivate them.”

It is unclear as to what the Authors refer to by this statement – their own results or initial preparative conditions of the WPC. In this form such statement may be confusing to the reader.

L30: it should be whey protein isolate (WPI) and whey protein hydrolysate (WPH)

L46: should be PFU not FPU

L74: in what aspect did these 15 WPCs differ? Where they from the same producer? Generated in the same/different dairy? What were the criteria of selecting these 15 particular WPCs? Some of these information can be found at various locations in the text, it would be profitable to group them and present in one place.

L76: it is unclear how many samples were analyzed from each WPC. Was each WPC analyzed against each of the 37 samples?

L79: was the turbidity test negative for WPC from dairy no. II? It is not mentioned at what stage these WPCs were determined to be free of phages. It could be that there were phages but there was no sensitive host among the 37 strains tested.

L87: it is unclear how phage purification steps were performed to ensure that the plaque corresponds to a single phage? Another issue: if the plaque was spread on the strain, then how was its MOI determined?

L89: how was the phage propagation carried out? Where the phages propagated from a single plaque? In liquid medium? On what strain?

L92: how was the PFU correlated to gram and not mL of phage lysate?? A specific formula how PFU was calculated per gram should be provided. I understand that gram refers to 1 gram of WPC, yet from the methodology description it seems that the phage concentration was not measured directly in the sample but after its consecutive propagation on provided Str. strain to increase phage concentration.

Table 1 - at what point was the phage concentration measured? Before or after enrichening the culture for phages?? This should be clearly stated.

The unit should be PFU not UFP

What is the Author's explanation for lack of phages in WPCs from dairy II? Lack of sensitive strain among the 37 Str. thermophilus tested?? Other type of WPCs used in this dairy??

The legend mentions WPC50, but in fact this type of WPC is not in the table or anywhere in the text, this should be explained. Perhaps it would be better to add the WPCs from dairy II to the table with the information that no phages were detected and data on its protein %.

L100: Spot test: was only a single spot used or was a dilution series produced? Using a single spot of high titer, does not rule out lysis from without which can result in false positives.

L100: How many phages were tested for their host range: 87? 83 or 73?? It seems from further descriptions that host range studies were done on 83 phages, which were not yet confirmed as distinct isolates. Would this not generate redundant data and prohibit proper interpretation of result data, both on Fig. 1 and Fig.2?

L104: here we read that phages were obtained at high titer – at what point this is described in section 2.2.?

L107: have these suspensions been previously concentrated (like was done in the cited reference) or was 1 mL of the high titer lysate taken for DNA isolation - this is not clear from the text

L120: what was the template in the PCR reaction - phage DNA with confirmed restriction pattern, lysate, plaque?? Was the same amount of template added to all reactions? This information can be crucial, e.g., in interpretation of the weak cos signal for the 5093 phage.

L128: should be ‘sec’ not ‘seg’

L138: how do the authors explain the difference between the stated phage titer 10^6PFU/mL vs the log PFU/mL shown on the graphs (Fig.7) - the initial titers indicated there are much higher than the stated value.

L141: how exactly were the titers determined? By double-layer plating of serial dilutions?? please provide this information

L150: should be ‘not’ instead of ‘no’

L151: it would be more informative to add that the WPCs M4, 11 and 12 derived from dairy II (e.g., to Table 1)

L155: 18, 15, and 17, according to table 1, are the number of phages detected in M13-15 not the number of sensitive strains, this discrepancy should be clarified

L162: it is confusing what the Authors mean by sample (WPC??) or isolate (phage with distinct restriction pattern?)

Figure 1 would be improved if an information stating what value is given in the bubble be provided in the figure caption, instead of searching it in the text.

L173: it is unclear how was this determined?? Before or after the three consecutive subcultures performed to increase phage concentration (as the Authors state in Mat and Met ); if the latter, then the determined titer does not correspond to the true/initial titer of the phages contaminating WPCs.

L184: again the same question - was the host range study performed on 73 distinct phages or the 83 isolates (which could give redundant results given the RFLP analyses)?

L187: what do the Authors mean by the control strain?? was it the same for all phages tested?

L192: what type of data is not shown? It is unclear what the Authors have in mind as they do not show any results of the phage infection profile other than the descriptive one.

Figure 2: would be improved if an information stating what value is given in the bubble be provided in the figure caption.

L201: it would be easier to follow the graph if strains would be divided as those sensitive against 1-5 and 6-20, 21-37 strains (or other range but allowing to read out the range from the graph)

Figure 3 would be improved if the strains on the graph will be arranged in the order of supplier (A, B, C and so on), then starter number (1,2,3, and so on) and finally strain letter.

L215: what is the reason for not obtaining 4 phage DNAs? Low phage titer?? Inability to propagate the phage on supplied strains? Or maybe the mentioned false positive read-out of spot tests could be misleading…One sentence explaining this would not leave the reader wondering.

L212: should be ‘infect each of them’ not ‘infect to each of them’

Figure 4: the horizontal axis should be explained in the figure legend.

There is sth wrong in the formatting of this figure, the right side of the figure is not visible.

Figure 5: leave out word scheme. The last sentence of the figure caption is rather a general statement and should be included in the main body text.

L242: would the weak band not indicate that the analyzed sample is a mix of phages - cos and 5093?? How were the plaques purified to assure that each derives from a single type of phage?? How many out of the 9 phages of 5093 showed such profile??

Figure 6 would be improved if control reactions for each phage genetic group were added. This could also convince the reader that the minor cos band for 5093 phage is not specific to all 5093 under the conditions of this test.

Thermal resistance: it is unclear what was the method of selecting the 5093 phages for thermal resistance assay or was it done randomly?? In Mat and Met it is stated that all pac phages were taken (as they were only 2), cos phages were selected based on either persistence in diverse samples or high virulence – a similar rationale for 5093 phages should be given?

L253: how was 5.8 log determined; for one phage the log reduction is much higher than 5.8 log

L278: in Mat and Met section it is stated that these experiments were performed in triplicates; how are the SD shown on these graphs?

Figure 8 would be improved if instead of 1,00 and 2,00 the name of each genetic group would be provided

L317: rather 15 to 18 isolates (not 9-18), acc to table 1

L320: is the M13 whey sample the only mixture of diverse origin; what about the other wheys? Is this indicated in the text somewhere??

Is there a study showing the relation between protein percentage in wheys and the number of surviving phages after thermal treatment? The Authors mention this in the text, but it would be profitable to provide a reference.

L350: in fact, we know only the samples derived from 3 origins, the reader cannot infer from the text were these different geographical regions

L361: is this conclusion made solely based on multiplex PCR genotyping?? How can the Authors prove it is not a contaminating phage that gives the cos signal?

L364: it would be more conclusive if host range profiles would be compared between the 73 phages regarded as distinct isolates and not isolates which could derive from the same phage

L367: here, like in Mat and Met section, it is difficult to infer what was the original host strain (or control strain in other places). Was this one of the 37 commercial Str strains or other?? how were these strains selected? If they were the only single strains (out of the 37) that were sensitive to these 4 phages, perhaps it is enough to say this, instead of introducing the term "original" strain or "control" strain, which provokes more questions than answers.

L382: there were only 2 pac phages isolated, so it should rather be stated that both pac phages had a wide host range

L391: it would be very valuable to this work to provide the data on the correlation of host range with genetic group in a table (or as supplementary data).

What do the Authors mean by additional? This suggests that some attempt to correlate the data was done, but this is not presented in the results section and only vaguely discussed in the discussion part.

L405: This conclusion is not supported by the provided data. With this line of thinking, one could say that pac phages as a well-described, relatively ‘old’, group of phages should be identified in high numbers, which is not the case.

L418: was this study conclusive in any way as to which phage suspension medium is the most protective for phage particles??

L445: What proof the Authors have for claiming that this group of phages is just emerging - any genetic proof, or sequenced-based???

How do the Authors plan to discuss the lack of phages in three out of 15 WPC - where these WPC rich (80) or less rich (35) in proteins?? Did they attempt to perform multiplex PCR directly on WPC samples?

Author Response

Response to Reviewer 2

The work describes the identification and basic characterization of Streptococcus thermophilus phages detected in whey protein concentrates. In a simple experimental approach, the Authors show that WPC can be an important cause of Streptococcus thermophilus phage infections in the Argentinian dairy environments. In an elegant way it is argued that whey protein concentrates used as additive in dairy fermentation may be rich in phage content. The work presents data on the infective properties and thermal resistance of 73 newly isolated Strep. thermophilus phages. By such approach the Authors aim to advance the description of the properties of this group of phages.

However, the manuscript demands some modifications and language edition.

  1. The introduction is concise and gives a nice overview on the use of whey derivatives in dairy fermentations and the consequences of their uncontrolled application. A minor comment to this part of the manuscript is that one could think of elaborating in more detail on the applied membrane technology in respect to ‘derivative type to be obtained’. Also to my knowledge microfiltration is also used besides ultrafiltration.

Several articles focused on use and obtaining process of diverse derivatives from cheese whey are available. We consider that it is not necessary to expand on this, since we wanted to limit our introduction to the main membrane technology for obtaining WPC.

  1. The materials and methods section lacks several information that would make the work plan easier to follow. The methodology is unclear at times as to how particular experiments were performed and on what type of samples (e.g., WPC or distinct phages) and by which method.

Materials and methods section was revised and modified according to the reviewer’s suggestions. Please, let us know if reviewer considers that additional changes are necessary or some details are missing.

  1. The results section launches into phage isolation from WPC and goes on to characterize their host range – but this was done before restriction pattern analyses which confirm the distinctness of each phage isolate. Should not it be that the Authors would first confirm that they are dealing with distinct, purified phages before analyzing their host range?? The Authors chose to refer to phage and strain names directly in the text rather than in tables – this does not allow to quickly associate the data to a specific phage/strain and makes the reader back-tracked at times (e.g., l162-164).

Host range and restriction pattern analyses were carried out at the same time. Based on the results of the two assays, similar/distinct phages were identified.

Text was modified in order to show the results in a clearer way (section 3.1). If reviewer consider additional changes are necessary, please, let us know.

  1. Discussion section relates the results of this study with literature data in respect to thermal resistance and host range and overall phage diversity among the isolates. However, some of the conclusions are not supported, while other aspects seem not to sufficiently discussed, e.g., it would add value to the publication to correlate the aspect of host range with genetic group, propose an explanation as to why no phages were detected in three WPCs all deriving from one source (dairy II), etc. It would be tempting also to try to correlate the protein % in WPC with the type of phages detected, also if possible it would be valuable to add an information on the genetic type of the detected phages in each WPC (i.e. in Table 1).

Relation between host range and genetic groups is well discussed in the manuscript (please, see lines 425-455).

Samples M4, M11 and M12, for which was not possible to isolate phages infective to the 37 St. thermophilus strains assayed, derived from diverse sources, that is: samples M4 and M11 came from whey-processing plant I, while sample M12 was provided by dairy II. Samples M4, M11 and M12 were included in the Table 1 in order to avoid misunderstandings.

A relation between protein percentage in WPC samples and the type of detected phage (e.g. genetic group) was not found in our work. Hence, we didn’t find it relevant in order to be included as information in Table 1.

  1. The conclusions mention the long-term aim of this work, being application of these phages to increase phage resistance of starter cultures. This is also stated in the title (application). However, little is written to describe what approach is envisioned – although one can imagine how this can be done, it still seems that this issue is described very vaguely, especially for it to be mentioned in the title of this manuscript.

Our research group have been studying CRIPR-Cas systems in St. thermophilus and Lactobacillus strains for several years. Once the CRISPR systems are characterized, our aim is to improve phage resistance of the strains used as starter cultures in the dairy industry, through CRISPR-Cas technology, by using the pool of new phages isolated and presented in this article. This was clarified in the conclusion of the manuscript.

Overall, the importance of the presented data has not been communicated in a manner that allows the reader to identify it and properly interpret and should be revised. Despite these major concerns there is lots of good things included in the manuscript that advance the characterization of Str. thermophilus phages. It is clear that much effort was put into production of this data which will contribute to a better understanding of the biology of these phages and their persistence in the dairy environment.

Specific comments

L14-15: “All phages survived to pasteurization treatments commonly applied in dairy industry being necessary, at least, 85 °C for 5 min to inactivate them.”

It is unclear as to what the Authors refer to by this statement – their own results or initial preparative conditions of the WPC. In this form such statement may be confusing to the reader.

We agree with the reviewer’s comment. Abstract of the manuscript was modified.

L30: it should be whey protein isolate (WPI) and whey protein hydrolysate (WPH)

We agree with the reviewer’s comment. Changes were done in the manuscript.

L46: should be PFU not FPU

We agree with the reviewer’s comment. FPU was replaced by PFU.

L74: in what aspect did these 15 WPCs differ? Where they from the same producer? Generated in the same/different dairy? What were the criteria of selecting these 15 particular WPCs? Some of these information can be found at various locations in the text, it would be profitable to group them and present in one place.

L76: it is unclear how many samples were analyzed from each WPC. Was each WPC analyzed against each of the 37 samples?

There was no specific selection criteria for WPC samples analyzed in this study. Independent samples were provided by plants/dairies in order to carry out a phage monitoring in their by-products. One sample of each WPC was analyzed against the 37 St. thermophilus strains.

Characteristics of these samples, as protein percentage and origin (producer) are detailed in Table 1. Negative samples (M4, M11 and M12) were also included.

L79: was the turbidity test negative for WPC from dairy no. II? It is not mentioned at what stage these WPCs were determined to be free of phages. It could be that there were phages but there was no sensitive host among the 37 strains tested.

As it was previously clarified, only one sample came from dairy II. For this sample and for two samples provided by plant I, no phages infective (negative turbidity test) on the assayed 37 St. thermophilus strains were detected.

These three samples are not considered as free of phages. This statement is impossible to ensure. Surely, samples contain phages able to infect to another St. thermophilus strains (or strains belonging to other species of lactic acid bacteria) not included in this study.

The selected commercial St. thermophilus strains were representative of strains used as starter cultures in Argentinian dairy industry, according to their diversity, suppliers and so on. However, it would be expected an increasing possibility of phage isolation by testing a higher number of strains.

L87: it is unclear how phage purification steps were performed to ensure that the plaque corresponds to a single phage? Another issue: if the plaque was spread on the strain, then how was its MOI determined?

L89: how was the phage propagation carried out? Where the phages propagated from a single plaque? In liquid medium? On what strain?

Phage purification by taking an isolated lysis plaque was performed. Then, it was spread on the respective sensitive strain until complete cell lysis. Afterwards, phage enumeration by double-layer plaque titration method was determined. Thus, we were able to calculate the m.o.i. for the subsequent phage propagation.

Phage propagation was conducted according to traditional (and routine) methodologies in liquid medium (M17 broth added of CaCl2). Each phage was propagated (and quantified through double-layer plaque titration method) on the strain with which they were isolated.

Methodology was clarified in the text (please, see lines 88-92).

L92: how was the PFU correlated to gram and not mL of phage lysate?? A specific formula how PFU was calculated per gram should be provided. I understand that gram refers to 1 gram of WPC, yet from the methodology description it seems that the phage concentration was not measured directly in the sample but after its consecutive propagation on provided Str. strain to increase phage concentration.

Phage enumeration (PFU per gram of WPC) was determined in the WPC samples. Titers of phages present in the WPC samples (reconstituted at 10% w/v) on each sensitive strain were determined and results were referred to gram of WPC (considering the previous dilution). These titers are shown in Table 1.

Propagated phages were also quantified in order to verify and assure high titers after propagation. These titers are not included in the manuscript.

Text was modified in the manuscript in order to clarify the methodology (please, see lines 89-96).

Table 1 - at what point was the phage concentration measured? Before or after enrichening the culture for phages?? This should be clearly stated.

The unit should be PFU not UFP

What is the Author's explanation for lack of phages in WPCs from dairy II? Lack of sensitive strain among the 37 Str. thermophilus tested?? Other type of WPCs used in this dairy??

The legend mentions WPC50, but in fact this type of WPC is not in the table or anywhere in the text, this should be explained. Perhaps it would be better to add the WPCs from dairy II to the table with the information that no phages were detected and data on its protein %.

As it was previously mentioned, phage concentration shown in Table 1 was determined in the WPC samples (not in the lysed and enriched cultures).

As it was previously explained, the impossibility of isolate phages from samples M4, M11 and M12 would be related to the absence of sensitive strains among the 37 St. thermophilus tested.

We are grateful for the reviewer’s suggestions in relation to Table 1. Table 1 was modified in order to show information in a clearer manner; negative samples were included and the mistakes were saved.

L100: Spot test: was only a single spot used or was a dilution series produced? Using a single spot of high titer, does not rule out lysis from without which can result in false positives.

Decimal dilutions were spotted onto the plates until to observe lysis plaques. This was clarified in the manuscript (please, line 110).

L100: How many phages were tested for their host range: 87? 83 or 73?? It seems from further descriptions that host range studies were done on 83 phages, which were not yet confirmed as distinct isolates. Would this not generate redundant data and prohibit proper interpretation of result data, both on Fig. 1 and Fig.2?

Eighty-three phages were tested for their host range (it was clarified in the manuscript, line 106). As it was previously described, host range, restriction pattern analyses and determination of genetic group were carried out at the same time. Similar/distinct phages were identified by analyzing the results of the three experiments. 

L104: here we read that phages were obtained at high titer – at what point this is described in section 2.2.?

Phage suspensions with high titer are obtained after their propagation.

L107: have these suspensions been previously concentrated (like was done in the cited reference) or was 1 mL of the high titer lysate taken for DNA isolation - this is not clear from the text.

Phage suspensions were not previously concentrated.  DNA was extracted from 1 mL of phage suspension with high titer (>108 PFU/mL), as it was described in the manuscript.

L120: what was the template in the PCR reaction - phage DNA with confirmed restriction pattern, lysate, plaque?? Was the same amount of template added to all reactions? This information can be crucial, e.g., in interpretation of the weak cos signal for the 5093 phage.

Extracted phage DNA was used in restriction enzyme analysis and genetic group determination.

The same amount of template (fluorometric measurement – Qubit) was added to all PCR reactions.

L128: should be ‘sec’ not ‘seg’

We agree with the reviewer’s comment and “seg” was replaced by “sec”.

L138: how do the authors explain the difference between the stated phage titer 10^6PFU/mL vs the log PFU/mL shown on the graphs (Fig.7) - the initial titers indicated there are much higher than the stated value.

We agree with the reviewer’s comment. Phage suspensions in titers of 106 - 108 PFU/mL were subjected to thermal resistance assays. This condition was added in the text. Phage concentration (of approximately 106 PFU/mL) was deleted from the text in order to avoid misunderstandings (section 2.3.4).

L141: how exactly were the titers determined? By double-layer plating of serial dilutions?? please provide this information

Yes. It was clarified in the manuscript.

L150: should be ‘not’ instead of ‘no’

We agree, text was modified.

L151: it would be more informative to add that the WPCs M4, 11 and 12 derived from dairy II (e.g., to Table 1)

Samples M4, M11 and M12 were included in Table 1.

L155: 18, 15, and 17, according to table 1, are the number of phages detected in M13-15 not the number of sensitive strains, this discrepancy should be clarified

L162: it is confusing what the Authors mean by sample (WPC??) or isolate (phage with distinct restriction pattern?)

We agree with the reviewer’s comment. Table 1 and section 3.1 were modified in order to avoid misunderstandings.

Figure 1 would be improved if an information stating what value is given in the bubble be provided in the figure caption, instead of searching it in the text.

Figure caption was modified.

L173: it is unclear how was this determined?? Before or after the three consecutive subcultures performed to increase phage concentration (as the Authors state in Mat and Met); if the latter, then the determined titer does not correspond to the true/initial titer of the phages contaminating WPCs.

It was previously explained. Titers in the WPC samples were determined.

L184: again the same question - was the host range study performed on 73 distinct phages or the 83 isolates (which could give redundant results given the RFLP analyses)?

It was previously explained.

L187: what do the Authors mean by the control strain?? was it the same for all phages tested?

The control strain was not the same for all phages. Strain with which phage was isolated was named/used as control strain.

It was clarified in the manuscript (please, see line 200).

L192: what type of data is not shown? It is unclear what the Authors have in mind as they do not show any results of the phage infection profile other than the descriptive one.

Data about the identification (name) of the most infective phages is not shown, although samples from which these phages were isolated are mentioned in the text. Specifically, results of host range showing the sensitivity of each strain with each phage is not included. We show a percentual distribution of these results in Figure 2.

“Data not shown” was deleted from the text in order to avoid misunderstandings.

Figure 2: would be improved if an information stating what value is given in the bubble be provided in the figure caption.

Figure caption was modified.

L201: it would be easier to follow the graph if strains would be divided as those sensitive against 1-5 and 6-20, 21-37 strains (or other range but allowing to read out the range from the graph)

We modified Figure 3 according to the following reviewer's suggestion. Therefore, if the graph is ordered in this way (which seems the most appropriate) we cannot modify it in the way suggested in this comment.

Figure 3 would be improved if the strains on the graph will be arranged in the order of supplier (A, B, C and so on), then starter number (1,2,3, and so on) and finally strain letter.

Figure 3 was modified according to the reviewer’s suggestion.

L215: what is the reason for not obtaining 4 phage DNAs? Low phage titer?? Inability to propagate the phage on supplied strains? Or maybe the mentioned false positive read-out of spot tests could be misleading…One sentence explaining this would not leave the reader wondering.

It was not possible to propagate these four phages. These four phages were not included in the molecular and host range studies.

A sentence was included in the manuscript (please, see lines 191-193).

L212: should be ‘infect each of them’ not ‘infect to each of them’

Correction was made in the manuscript.

Figure 4: the horizontal axis should be explained in the figure legend.

There is sth wrong in the formatting of this figure, the right side of the figure is not visible.

Figures 4 and 5 were modified.

We think it is not relevant to include the phage name in the Figure (each line corresponds to a phage). However, if reviewer considers necessary to include them, please let us know.

Figure 5: leave out word scheme. The last sentence of the figure caption is rather a general statement and should be included in the main body text.

We don’t find the word scheme in the manuscript.

We don’t understand the reviewer’s comment about the last sentence of the figure caption. Information showed in the figure caption is also included in the materials and methods section.

L242: would the weak band not indicate that the analyzed sample is a mix of phages - cos and 5093?? How were the plaques purified to assure that each derives from a single type of phage?? How many out of the 9 phages of 5093 showed such profile??

Two phages out of 9 belonging to group 5093 evidenced a weak cos band.

DNA of purified phages was extracted. Assays were repeated in order to confirm the results. We plan to study it in more detail, beginning with the sequencing of these phages.

Figure 6 would be improved if control reactions for each phage genetic group were added. This could also convince the reader that the minor cos band for 5093 phage is not specific to all 5093 under the conditions of this test.

Control reactions for phages cos and pac (belonging to INLAIN collection) were included in our experiments (but they are not shown in the manuscript). In that moment, we had not phages belonging to groups 5093 and 987 to be used as reaction control. Genetic group is determined considering the size of the expected amplified product.

Thermal resistance: it is unclear what was the method of selecting the 5093 phages for thermal resistance assay or was it done randomly?? In Mat and Met it is stated that all pac phages were taken (as they were only 2), cos phages were selected based on either persistence in diverse samples or high virulence – a similar rationale for 5093 phages should be given?

Seven phages out of a total of nine isolated belonging to 5093 group were included in the thermal characterization. The remaining two phages were not included in the study of this phenotype because they showed difficulty in plaque formation under the titration conditions. It was clarified in the manuscript (please, lines 145-146).

L253: how was 5.8 log determined; for one phage the log reduction is much higher than 5.8 log

We agree with the reviewer’s comment. The log reduction was, at least, of 5.8 for the two pac phages. Text was modified.

L278: in Mat and Met section it is stated that these experiments were performed in triplicates; how are the SD shown on these graphs?

Thermal resistance assays were performed in triplicate. Figures 7 with (and without) error bars were included in the manuscript. We think it would look better if figure 7 without error bars is included. Standard deviations of titer reductions caused by thermal treatment are included in Table 3. Maybe it is more interesting to show standard deviations of titer reductions than those for initial and final titers.

Figure 8 would be improved if instead of 1,00 and 2,00 the name of each genetic group would be provided

We agree with the reviewer’s suggestion. Figure 8 was modified.

L317: rather 15 to 18 isolates (not 9-18), acc to table 1

We agree with the reviewer’s comment. Text was modified.

L320: is the M13 whey sample the only mixture of diverse origin; what about the other wheys? Is this indicated in the text somewhere??

Is there a study showing the relation between protein percentage in wheys and the number of surviving phages after thermal treatment? The Authors mention this in the text, but it would be profitable to provide a reference.

L350: in fact, we know only the samples derived from 3 origins, the reader cannot infer from the text were these different geographical regions

All samples provided by whey-processing plant I were obtained from a mixture cheese whey of diverse origins (many Argentinian dairies). From the three most infective samples, only sample (M13) came from plant I. The remaining 2 samples were provided by dairies II and III. Dairy II and III process their own cheese whey in order to produce WPC.

A sentence to clarify this issue was added in the manuscript (please, see lines 75-76).

Studies relating protein content of the whey and thermal resistance of the phages were discussed in the manuscript (please, see lines 481-493).

L361: is this conclusion made solely based on multiplex PCR genotyping?? How can the Authors prove it is not a contaminating phage that gives the cos signal?

This is a hypothesis, as it is mentioned in the manuscript. We will investigate this hypothesis, beginning with the phage sequencing.

L364: it would be more conclusive if host range profiles would be compared between the 73 phages regarded as distinct isolates and not isolates which could derive from the same phage

It was previously clarified. Host range and restriction pattern analyses were carried out at the same time. Similar/distinct phages were identified by analyzing the results of the two experiments. 

L367: here, like in Mat and Met section, it is difficult to infer what was the original host strain (or control strain in other places). Was this one of the 37 commercial Str strains or other?? how were these strains selected? If they were the only single strains (out of the 37) that were sensitive to these 4 phages, perhaps it is enough to say this, instead of introducing the term "original" strain or "control" strain, which provokes more questions than answers.

The control strain was not the same for all phages. Strain with which phage was isolated was named as control strain.

Term “original” was deleted from the text.

L382: there were only 2 pac phages isolated, so it should rather be stated that both pac phages had a wide host range

We are in agreement with the reviewer’s comment. Text was clarified.

L391: it would be very valuable to this work to provide the data on the correlation of host range with genetic group in a table (or as supplementary data).

What do the Authors mean by additional? This suggests that some attempt to correlate the data was done, but this is not presented in the results section and only vaguely discussed in the discussion part.

We consider information about correlation between host range and genetic group is enough and it is clearly showed/discussed in the manuscript. We think it would be not necessary to provide data in a table, as it is suggested by the reviewer.

It would be interesting to conduct additional studies relating genetic group with host range in order to confirm the described hypothesis (e.g. pac phages are more infective than cos phages) since only two pac phages were isolated in our work. For this purpose, isolation (and subsequent characterization) of a greater number of pac phages would be necessary.

L405: This conclusion is not supported by the provided data. With this line of thinking, one could say that pac phages as a well-described, relatively ‘old’, group of phages should be identified in high numbers, which is not the case.

To our knowledge, the literature indicates that cos phages are isolated in greater numbers than pac ones, but the reason of this fact is unknown to us.

L418: was this study conclusive in any way as to which phage suspension medium is the most protective for phage particles??

Our study was performed in TMG buffer in order to know the intrinsic thermal resistance of the phages, then establishing a relationship with the genetic group. Nowadays, research about protective effect of whey and WPC on phage particles thermally treated are in progress.

L445: What proof the Authors have for claiming that this group of phages is just emerging - any genetic proof, or sequenced-based???

Genetic groups 5093 and 987 are named “emerging” by some authors (Mc Donnell et al., 2016; Mc Donnell et al., 2017; Philippe et al., 2020)*. These are recently identified groups, based on the phage sequencing.

“Emerging” was replaced by “identified recently group” in the manuscript.

* McDonnell, B.; Mahony, J.; Neve, H.; Hanemaaijer, L.; Noben, J.P.; Kouwen, T.; van Sinderen, D. Identification and analysis of a novel group of bacteriophages infecting the lactic acid bacterium Streptococcus thermophilus. Appl. Environ. Microbiol. 2016, 82, 5153–5165.

McDonnell, B.; Mahony, J.; Hanemaaijer, L.; Neve, H.; Noben, J.P.; Lugli, G.A.; Ventura, M.; Kouwen, T.R.; van Sinderen, D. Global survey and genome exploration of bacteriophages infecting the lactic acid bacterium Streptococcus thermophilus. Front. Microbiol. 2017, 8, 1–15.

Philippe, C.; Levesque, S.; Dion, MB.; Tremblay, DM.; Horvath, P.; Lüth, N.; Cambillau, C.; Franz, C.; Neve, H.; Fremaux, C.; Heller, KJ.; Moineau, S. Novel genus of phages infecting Streptococcus thermophilus: genomic and morphological characterization. Appl. Environ. Microbiol. 2020, 86, 1–15.

How do the Authors plan to discuss the lack of phages in three out of 15 WPC - where these WPC rich (80) or less rich (35) in proteins?? Did they attempt to perform multiplex PCR directly on WPC samples?

As it was previously mentioned, samples could contain phages able to infect to another St. thermophilus strains (or strains belonging to other species of lactic acid bacteria) not included in the study. No multiplex PCR was directly applied on WPC samples (these assays only detect DNA but do not provide information about the presence of infective phages).